# SimTalk: Simulation of IoT Applications

**DOI:** 10.3390/s20092563

**Published:** 2020-04-30

**Authors:** Yun-Wei Lin, Yi-Bing Lin, Tai-Hsiang Yen

**Affiliations:** 1College of Artificial Intelligence, National Chiao Tung University (NCTU), Hsinchu 300, Taiwan; jyneda@nctu.edu.tw; 2Department of Computer Science, National Chiao Tung University (NCTU), Hsinchu 300, Taiwan; ksoy.cs08g@nctu.edu.tw

**Keywords:** simulation, IoT, sensor, actuator, smart farm, interactive art, physics experiments

## Abstract

The correct implementation and behavior of Internet of Things (IoT) applications are seldom investigated in the literature. This paper shows how the simulation mechanism can be integrated well into an IoT application development platform for correct implementation and behavior investigation. We use an IoT application development platform called IoTtalk as an example to describe how the simulation mechanism called SimTalk can be built into this IoT platform. We first elaborate on how to implement the simulator for an input IoT device (a sensor). Then we describe how an output IoT device (an actuator) can be simulated by an animated simulator. We use a smart farm application to show how the simulated sensors are used for correct implementation. We use applications including interactive art (skeleton art and water dance) and the pendulum physics experiment as examples to illustrate how IoT application behavior investigation can be achieved in SimTalk. As the main outcome of this paper, the SimTalk simulation codes can be directly reused for real IoT applications. Furthermore, SimTalk is integrated well with an IoT application verification tool in order to formally verify the IoT application configuration. Such features have not been found in any IoT simulators in the world.

## 1. Introduction

Many smart applications have been developed with Internet of Things (IoT) technology, including home automation [1], a smart aquarium [2,3], an intelligent campus [4,5], precision agriculture [6,7], interactive art and entertainment [8,9,10], and more. However, it is seldom mentioned how they are correctly implemented, especially for the existing or envisioned applications in remote sensing. To develop these applications, simulations provide a cost-effective verification approach to end-to-end execution. Through simulation, we can also evaluate the conceivability of applying particular techniques to the target IoT applications, which shed light on directions for possible future implementation. We will elaborate on existing IoT simulation solutions in Section 2. These solutions provide guidelines for implementing the real IoT applications. However, the simulation codes of these solutions cannot be directly reused for real applications due to their “discrete event” nature. Furthermore, since the codes of the real applications are separately developed, it does not guarantee that the codes for real IoT applications are consistent with the simulation codes. An extra verification process is required. 

To resolve this issue, this paper proposes SimTalk based on an IoT development platform called IoTtalk [11]. The underlying concept of SimTalk is to develop a time-driven simulation that can automatically translate the simulation codes to those for the real IoT applications and vice versa. Therefore, when we complete the simulation, the codes can be automatically translated into real IoT applications running on IoTtalk. 

IoTtalk offers a graphical user interface (GUI) to describe the relationship between sensors and actuators graphically, allowing simple data manipulation and transfer between IoT devices to occur. After an IoT service has been developed, IoTtalk GUI binds the network application program with the real sensors and the actuators with a button click. Since the IoTtalk application programs are implemented in Python, the IoT service is immediately provisioned without compilation when the real IoT devices are bound to the network application program.

In this paper, we extend IoTtalk with the simulation functionality by providing a simulation soft switch in the IoTtalk GUI. Before the real IoT devices are bound to the IoT applications, the developer can use SimTalk to bind the simulated sensor corresponding to a real sensor, set up specific traffic patterns to the simulated sensor to drive the IoT service, and then observe its behavior for correctness and function improvement. The paper is organized as follows. Section 2 surveys the related IoT simulation solutions. Section 3 provides an overview to the IoTtalk architecture. Section 4 proposes the SimTalk architecture based on IoTtalk. Section 5 discusses the details of SimTalk implementation for the sensors. Section 6 elaborates on the simulators for the SimTalk actuators.

## 2. Related Work

A good survey of IoT simulation is given in [12], which compared the state-of-the-art IoT simulation solutions. This survey points out that an IoT research project typically consists of two phases.

Phase 1. A proof-of-concept is realized in the virtual domain using simulation.

Phase 2. The real IoT application is implemented and experimented on a testbed.

Many simulation studies address Phase 1 in [12]. One of these studies [13] proposed agent-based adaptive parallel discrete-event simulation to enhance scalability and to permit simulation of largescale smart cities. In [14], the authors proposed YAFS (Yet Another Fog Simulator), a discrete-event fog computing simulator to model the relationships among the real IoT applications, network connections and infrastructure characteristics. In particular, the authors modeled three scenarios for dynamic allocation of new application modules, dynamic failures of network nodes and user mobility along the topology. IOTSim [15] is a simulator that enables simulation of IoT big data processing using MapReduce model in a cloud computing environment. In [16], the authors proposed an agent-oriented approach for modeling IoT networks. Specifically, by using the Omnet++ simulation platform, the ACOSO model is exploited to simulate IoT networks of various scales. Then the simulation results are used to analyze the bottlenecks at communication level. In [17], the authors also used Omnet++ to simulate the IoT applications with hardware in the loop. The resulting discrete-event simulation increases quality and significance of the modeling results, and enables the analysis of components that are not available at early stages of the development cycle.

The state-of-the-art simulation solutions described above live up to the goal of Phase 1, and the simulation results provide very good guidelines to implement the real IoT applications in Phase 2. However, the simulation codes of these solutions cannot be directly reused for real applications due to the “discrete event” nature of the codes (where the clock of the simulation is advanced by an event scheduler based on the timestamps of the events, which is not found in the real applications). Furthermore, since the codes of the real applications are separately developed in Phase 2, it does not guarantee that the codes in Phase 2 are consistent with the simulation codes in Phase 1. An extra verification process is required.

Unlike the above discrete-event simulation solutions, SimTalk follows the time-driven simulation approach that does not need any event scheduler used in an event-driven simulation, and can be implemented in a real IoT development environment (like IoTtalk) by using the real time advancing mechanism. With SimTalk, the development of an IoT application project seamlessly advances from Phase 1 to Phase 2 by simply changing the execution time units. There is no need to verify if the codes in both phases are consistent because they are the same in SimTalk.

Besides simulation capability, SimTalk and IoTtalk are nicely integrated with two tools that verify the correctness of SimTalk and sensor data accuracy of IoTtalk. Such features are not found in any IoT simulators in the world. Based on the theory of bigraphs [18], correct configurations of IoT devices in SimTalk are formally verified by BigraphTalk [19], a verification framework that utilizes formal techniques to statically guarantee that there are no unwanted sensor-actuator configurations. BigraphTalk checks for invalid connections between devices, as well as type errors, e.g., passing a float to a Boolean switch. In [20,21], methods have been proposed to guarantee that the sensor data are correct. In particular, the SensorTalk approach [20] has been integrated with IoTtalk to automatically detect potential sensor failures and calibrates the aging sensors semi-automatically. When the sensors trigger the actuators, SensorTalk can detect failures within a short detection delay so that when a potential failure occurs, it is detected reasonably early without incurring too many false alarms. Details of BigraphTalk and SensorTalk are out of the scope of this paper, and can be found in [19,20].

## 3. The IoTtalk Architecture

As an IoT application development platform, IoTtalk is defined in two domains [11]. In the device domain, an IoTtalk device (such as a PM2.5 sensor or a light actuator) consists of two software components:The Sensor and Actuator Application (SA; Figure 1a) is responsible for the implementation of the IoT device function such as the PM2.5 algorithm or the light intensity and color circuit software.The Device Application (DA; Figure 1b) is responsible for communications with the IoTtalk network domain. The communication technique can be wired or wireless (e.g., WiFi, LTE, NB-IoT, RoLA, and more [22]).

The DA/SA software of an IoT device can be automatically generated (AG; Figure 1a,b) or manually created (Figure 1c).

In the network domain, an IoTtalk server is responsible for provisioning the network applications that manipulate the IoTtalk devices. An IoTtalk service or project (such as smart home or smart agriculture) is a set of network applications. The server consists of several subsystems (Figure 1d–k). The Execution and Control Subsystem (EC; Figure 1d) is responsible for the control plane (the Control submodule) and the user plane (the Execution submodule) of the end-to-end path between the IoTtalk devices and the server. The Creation, Configuration and Management (CCM; Figure 1e) subsystem systematically creates and manages the network applications of the IoTtalk devices for the corresponding IoT services. IoTtalk defines a device model for real devices with the same properties (see Appendix A for the details). For example, a smartphone device model is mapped to various real smartphones such as iPhone, iPad, Android smartphones and more. A device model consists of several device features (DFs). For example, the acceleration sensor of a smartphone is an input DF (IDF) that sends data to the EC, and the speaker of the smartphone is an output DF (ODF) that receives instructions from the EC. For the configuration purpose, we further partition a device model into the input and the output parts. The input device model is the set of IDFs (e.g., the acceleration sensor, the gyro sensor, the camera and the soft keys of a smartphone) and the output device model is the set of ODFs (e.g., the speaker and the screen of the smartphone). The CCM is responsible for managing the device models and their DFs, and stores such information in the IoTtalk database (DB; Figure 1f). The IoTtalk GUI (Figure 1g) is a friendly web-based user interface that allows a developer to quickly establish the connections and meaningful interactions among the IoT devices. The Authentication, Authorization and Accounting Subsystem (AAA; Figure 1h) is responsible for the management of user accounts and access to the IoTtalk applications. Figure 1d–h are core components in a typical IoT platform, where the developer manually creates IoT devices and IoT services through the IoTtalk GUI. Two major network protocols are used in IoTtalk: Message Queueing Telemetry Transport (MQTT) are used in the links (b)–(d), (c)–(d), (d)–(e) and (e)–(g). HTTPS is used in the link (e)–(g). The DB interacts with the CCM through the ORM protocol.

Figure 2 illustrates a smart farm service where the farmer uses the weather station (Figure 2a) and the timers (Figure 2b) to control the farming actuators such as the lights (Figure 2c) and the sprayers (Figure 2d).

Figure 3 shows how this simplified smart farm service is created by using the IoTtalk GUI (Figure 1g). From the model pulldown menu (Figure 3a), we select two input device models: the Sensors model (Figure 3b) implements the DA/SA for the weather station and the Timers model (Figure 3c) implements the DA/SA for multiple timers. Similarly, we select the output device model actuators (Figure 3d) that implement the DA/SA for the farming actuators. The Sensors model includes Lum-I (the IDF for the luminance sensor) and Hum-I (the IDF for the humidity sensor). The Timers model includes two IDFs for two timers. The Actuators model has Light-O (the ODF for the lights) and Spray-O (the ODF for the sprayers). To control the lights by the luminance sensor, we simply drag a line to connect Lum-I and Light-O in the GUI. In this simple smart farm service, the lights are controlled by both the luminance sensor and timer 1 through the Join 1 link. Similarly, the sprayers are controlled by both the humidity sensor and timer 2 through the Join 2 link.

By clicking the upper-right corner of the Sensors icon (Figure 3e), the SA/DA of the Sensors model is bound to the real device, i.e., the weather station in Figure 2a, and the service is activated for execution.

## 4. The SimTalk Architecture

IoTtalk also provides an advanced feature called AutoGen, which can automatically generate devices and projects (services). An example is EduTalk [23] that is a physics and Python programming course platform. We use AutoGen to automatically create an IoTtalk project for an EduTalk course lecture to perform physics experiment through interaction with a smartphone. AutoGen is also used to interwork IoTtalk with any NB-IoT systems, where IoTtalk automatically creates a device called NB-IoTtalk for every NB-IoT service. This NB-IoTtalk device provides interaction between the NB-IoT devices (e.g., the parking sensors) with any existing IoTtalk devices to build new services [22]. Like NB-IoTtalk, AutoGen is used to interwork IoTtalk with various AI tools by packaging these AI tools as IoT devices derived from the ML_device device model [24,25]. This novel approach for integrating IoT with “X” systems is provided by the AutoGen Subsystem (Figure 1i) that is considered as a platform to create “X-Talk” Subsystems (Figure 1j). In the current IoTtalk version, X = Edu (education), NB-IoT, and Artificial Intelligence (AI). Every X-Talk service is associated with a web-based GUI that allows the developer to set up the parameters of the automatically generated device. For example, the developer uses AItalk GUI to select the machine-learning algorithms such as support vector machine (SVM), k Nearest Neighbor (kNN), Decision Tree, Random Forest, and so on. After the developer has set up the device parameters, the AutoGen Subsystem creates the device (Figure 1a,b). In this paper, we will focus on SimTalk, the simulation capability for IoTtalk, which is built on top of the AutoGen Subsystem. The SimTalk Subsystem (Figure 1j) is associated with a web-based GUI (Figure 1k) that allows the developer to set up the parameters of the automatically generated device. In Figure 1, HTTPS is used for (e)–(i) and (j)–(k). The arrow link (g)->(k) represents page jumps from the IoTtalk GUI to the X-Talk GUI.

The idea of SimTalk is described as follows. To further explore the AutoGen feature, for every real input device in a project, we can automatically create a counterpart simulated sensor to faithfully simulate the behavior of that input device. The detailed functional block diagram of the SimTalk and the AutoGen Subsystems in Figure 1 are illustrated in Figure 4. In this figure, the SimTalk GUI (Figure 4a) allows the developer to set up the parameters of the simulated sensors. The SimTalk Event Handler (Figure 4b) receives the instructions from the SimTalk GUI and the AutoGen Subsystem (Figure 4c), and executes the Simulator Management Procedures (Figure 4d) corresponding to these instructions. The execution results are saved in the SimTalk DB (Figure 4e). In the AutoGen Subsystem, the AutoGen Event Handler (Figure 4f) receives the instructions from the SimTalk Subsystem and the CCM (Figure 4g), and executes the AutoGen Management Procedures (Figure 4h) corresponding to these instructions.

We note that the AutoGen Procedures are generic that apply to all X-Talks including EduTalk, NB-IoTtalk, AItalk and SimTalk.

## 5. Implementing SimTalk

When the IoTtalk server is installed, all databases including the SimTalk database (DB; Figure 4e) are initiated. When the AutoGen Subsystem is initiated, it executes the initiation program (Figure 4i) to find the host of SimTalk Subsystem in the X-Talk Configuration File (Figure 4j). This host information is used to invoke the SimTalk initialization program (Figure 4k) that creates the threads for the SimTalk GUI and the SimTalk Event Handler.

We use the Sensors device model as an example to show how to set up a simulated sensor. By clicking the gear icon in the upper-left corner of Sensors (Figure 3b), the Sensors DF setup table pops up (Figure 3f). When we click the “Extra Setup” button (Figure 3g), the IoTtalk GUI (Figure 4l) jumps to the SimTalk GUI (Figure 4a). The SimTalk GUI sends a query request to the SimTalk Event Handler. The handler invokes the Query Parameters procedure (Figure 4m) to send the request to the AutoGen Event Handler. Through the X-Talk HTTP Service procedure (Figure 4n), the event handler forwards the query to the CCM (Figure 4g). The CCM retrieves the parameters of IDFs (i.e., Lum-I and Hum-I) from the IoT DB (Figure 1f), and returns them to the SimTalk Subsystem through the AutoGen Subsystem. The SimTalk Event Handler invokes the Save Parameters procedure (Figure 4o) to create a record for Sensors in the SimTalk DB (Figure 4e), and saves all IDF parameters in this record.

The handler returns the Sensors record to the SimTalk GUI. This record is used to create a parameter setup window for the Sensors device model (Figure 5a), which lists Lum-I and Hum-I to be simulated. In this window, when the Lum-I button (Figure 5b) is clicked, the luminance simulated sensor setup window pops up. In this window, we set up the inter-arrival times by selecting the distribution (fixed, Exponential, Gamma, and more) through the corresponding pull down menu (Figure 5c). Similarly, we determine the variance and the mean of the distribution through the buttons in Figure 5d,e. If all IDFs of Sensors have the same inter-arrival times, then we can set up it by clicking the Sensors button (Figure 5a) with the similar setup procedure described above.

We also need to set up the parameters for IDF value generation, including its distribution (Figure 5g), the mean (Figure 5h) and the variance (Figure 5i). We note that the values of some IDFs have ranges. For example, the relative luminance ranges from 0 to 128,000. For this kind of IDF, the simulated distributions must be finite or truncated to produce the values in the ranges restricted by the upper and the lower bounds. In this case, we need to fill the lower bound value (Figure 5j) and the upper bound value (Figure 5k).

To reproduce the same simulation sequence, the SimTalk GUI allows the user to specify the seeds for random number generation through the fields “Seed” (Figure 5f,l). The “Trace File” field (Figure 5m) allows the user to download a recorded trace to conduct trace-driven simulation. Note that IoTtalk records all sensor data of a real IoT application with timestamps in specified periods and saved them in a trace file. Therefore, the user can upload the trace file through the “Trace File” field. We can also perform distribution-fitting that translates the measured data in the trace file to a Gamma distribution [26]. In [3,6,10,11,22], we have manually conducted distribution fitting to make the simulation cases more valid. Implementation of automatic distribution-fitting in SimTalk will be our future work.

After the IDF setup is finished, we click the “Save” button (Figure 5n) to close the window. The SimTalk GUI passes the setup values to the SimTalk Event Handler, and the handler invokes the Save Parameters procedure to save these values into the Sensors record in the SimTalk DB.

When the user clicks the “Simulation button” in the IoTtalk GUI window (Figure 6a), a page jump occurs and the SimTalk GUI sends the “create simulators” instruction to the SimTalk Event Handler. The handler forwards the request to the CCM through the AutoGen Subsystem. The CCM queries the IoTtalk DB to retrieve the IDF parameter setups of all input devices in the SmartFarm project. These parameter values are returned to the AutoGen Event Handler. By invoking the X-Talk HTTP Services procedure, a response with these values is sent to the SimTalk Event Handler. The handler saves the parameter values in the SimTalk DB and invokes the Query Parameters procedure to send a response with these parameter values to the SimTalk GUI.

Based on the received parameter values, the GUI pops up a window illustrated in Figure 7 to show the status of each input device model. For SmartFarm, they are Sensors and Timers. The window indicates that the Sensors device model is currently bound to a real device (Figure 7a). Therefore, the user can choose to keep the binding or unbind the real device and switch Sensors to simulation mode by clicking the simulator radio button (Figure 7b). The Timers device model is not bound to any real device, and the user did not set up the simulator parameters. Therefore, this device model is always bound to a simulated device with default parameter values (Figure 7c).

Note that the user can fill multiple values in each of the parameter text boxes (Figure 5d,h,e,i), and then the SimTalk simulation will sweep on these values. Each set of the values is executed for an observation period (Figure 7d). This feature is called parameter sweeping. If the parameter sweeping mode is not selected (Figure 7f), then the simulation is only executed for the first set of the values. We have designed the parameter sweeping feature for SimTalk and the implementation will be our future work.

When the user clicks the “Save” button (Figure 7g), the SimTalk GUI sends the SimTalk Event Handler a “save” request with a list of the input device model names selected as simulators. The handler saves this list in the SimTalk DB, invokes the Create Simulator procedure (Figure 4p) to generate the SA code of each simulated device (see Appendix B for the details) and requests the AutoGen Subsystem to create the simulator. The AutoGen Event Handler invokes the Create Device procedure (Figure 4q) to create the devices for the simulator and sends a “bind device” request to the CCM. The CCM binds these simulated devices to the SmartFarm project and the simulation of SmartFarm is started. A successful response is sent from the CCM to the SimTalk GUI. If the simulation fails, the GUI pops up a dialog window to indicate the simulation status.

The SimTalk Subsystem reuses all IoT device codes in the IoTtalk application and inserts random number generators into the codes with the sleep() function. Furthermore, the interactions between the input devices and the output devices in SimTalk are specified in the IoTtalk GUI (Figure 3). In this way, we guarantee that the IoT device interaction in SimTalk is exactly the same as that of the real applications.

The created simulation code will be executed to conduct time-driven simulation by advancing the real clock. That is, the time advancing mechanism of SimTalk simply follows the same mechanism of IoTtalk. The progress of time in SimTalk is specified in the time interval field (Figure 5c–f), and the clock of each simulated IoT device is incremented with the time units (specified in Figure 7e) through advancing time of the real execution of IoTtalk. Therefore, SimTalk does not need any event scheduler used in the event-driven simulation. In IoTtalk, the real clock is advanced by the sleep() function in a sensor device. For example, if a temperature sensor periodically sends the measured data every 10 min, then the temperature code simply calls sleep (600) in seconds. If we change the time unit from “minute” to “second” then SimTalk can simulate the application 60 times faster than the execution of the real application in IoTtalk. That is, the unit for the time intervals can be set up so that the progress of time is much faster than the real device execution. In SimTalk emulation, the time intervals of the simulated IoT devices must be specified exactly the same as its real device counterparts. As another example, in our work on Elevator simulation and emulation [27,28], a real elevator car takes 3.46 s to move up/down one floor. In the simulation, the movement of all simulated cars are delayed with the sleep() function for 0.00346 s. In the emulation, sleep() must delay for exactly 3.46 s to synchronize the simulated elevator cars with the real cars.

In a SimTalk application, if all IoT devices are simulated devices, then SimTalk conducts the simulation. If some IoT devices are real, and the remaining devices are simulated (i.e., the real devices are mixed with the simulated devices), then SimTalk conducts emulation. If all IoT devices are real, then SimTalk become IoTtalk to run real IoT applications. To switch between SimTalk and IoTtalk, one simply clicks the “Simulation” button (Figure 6a).

In the time-driven simulation, special treatments are required to simulate the sensors with very different frequencies. Suppose that we simulate *N* sensors with the sampling periods $t_{n},\;1\leq n\leq N$, where the maximal period is $t_{M}$ and the minimum period is $t_{m}.$ Then the time unit for sleep() is selected such that the central processing unit (CPU) speed of SimTalk is faster than the rate to handle the $s_{m}$ data at the frequency higher than $1/E[t_{m}]$. If the variances of the $t_{n}$ distributions are not large, then the time complexity of the execution for the time-driven simulation is about the same as the discrete-event simulation. On the other hand, if the variance of $t_{m}$ is very large, then it is possible that within two samples of $s_{M}$, there is only one $s_{m}$ sample or more than 1000 $s_{m}$ samples. In the former case, SimTalk’s CPU will loop in $s_{M}$’s sleep() function for a long time. Therefore, in terms of the execution time, the event-driven approach is typically more efficient than time-driven if the events occur with high variance. Fortunately, in most IoT applications we encountered, the sensors produce data with the intervals on the same order. In the future, we will investigate the IoT applications where the events occur with very high variance, and to study how SimTalk can be modified to effectively simulate such applications.

## 6. Simulating Output Devices

In Section 3, Section 4 and Section 5, we elaborated on how to create the simulator for the input devices. Basically, these simulated devices are traffic generators that drive the IoT services for the correctness investigation. In particular, simulation results can provide useful suggestions to design physical devices for IoT field trials. For an output device model, it is important that the simulation provides two types of information:The input values sent from the EC to the ODFs of the simulated output device; andThe results produced by the output device.

To produce the first type of information, SimTalk utilizes the “ODF monitor” in the IoTtalk GUI (Figure 6b). This monitor produces the values (Figure 6c) and their timestamps (Figure 6d) received by the ODF. The ODF monitor is a good mechanism for debugging. Besides the raw data shown in Figure 2c,d, the data can also be illustrated in the statistics charts as shown in Figure 8.

To investigate the second-type information, SimTalk also provides an animated simulator for the output device models. An example is the skeleton ceiling light that can change shape and light color during the night and reflect the shadow of different geometric shapes on the floor (Figure 9a). The shape change is achieved through compression of the skeleton stalks with various angles and sizes. Therefore, the skeleton device model has three ODFs: Angle-O, Size-O and Color-O. The skeleton simulator is an animation program that reuses SA of its physical counterpart to illustrate compression of the stalks of the skeleton with color change. The animated skeleton simulator is implemented in Java, which continuously draws the graphical skeleton patterns with specified ODF values (Figure 9b). The implementation details are given in [5].

The skeleton device is connected to a web-based controller with three IDFs. Every IDF is a 3 × 3 keypad. Color-I is a 9-color palette (Figure 10c). Angle-I is a 9-number keypad (Figure 10d) that produces an angle of a decimal degree n when the *n*-th key is pressed, where *n* = 10, 20, …,90. Size-I is another 9-number keypad that produces a size m when the m-th key is pressed, where 1 ≤ m ≤ 9. The skeleton configuration created through the IoTtalk GUI is illustrated in the right-hand side of Figure 10. A user accesses the controller (Figure 10a) through the browser of his/her smartphone. Then he/she can press the keypads to interact with the skeleton device (Figure 10b).

When the controller is switched to the simulation mode, the skeleton is driven by random sequences of the angle, the color, and the size. In Figure 10, we can add another skeleton device model (called Skeleton2) bound to a newly built Skeleton C as illustrated in Figure 9c. To ensure that Skeleton C correctly duplicates Skeleton A in Figure 9a, the simulated controller connected to Figure 9b is also connected to Skeleton2. When the simulation (actually, emulation) executes the input sequences produced by the simulated controller, we can observe if Skeleton C has the same behavior as the simulated Skeleton (Figure 9b). If so, Skeleton C has the same behavior as Skeleton A without any operation on Skeleton A.

The results of output device simulation can provide useful suggestions to design physical devices for IoT field trials. For example, in National Chiao Tung University (NCTU), the IoTtalk platform is used to implement the water dance service in a fountain with the sprinklers and lights. Figure 11 shows that 6 simulated light bulbs are mapped to the lights under the water of the fountain. There are also 6 sprinklers and their simulated counterparts. To simplify our discussion, they are not shown. After the water-dance designer is satisfied about the control sequences of the lights and the sprinklers through the simulation (typically controlled by the music), he/she can bind the real lights and sprinklers through the IoTtalk GUI.

We can also use simulated output devices to investigate the behavior of an input device. This feature is especially useful for physics experiments. Figure 12 shows how the behavior of a pendulum (Figure 12a) can be observed by its simulation counterpart (Figure 12b).

In Figure 12, in the bob of the pendulum is installed an acceleration sensor and a WiFi/Bluetooth wireless module. The SA of the pendulum implements the IDF Acceleration-I. When the bob swings back and forth, the bob sends the acceleration values to the IoTtalk server through WiFi or Bluetooth. The IoTtalk server then links Acceleration-I to the corresponding ODF of an output simulator that is a 2-D pendulum animation written in VPython. The SA of the simulated pendulum uses the acceleration data to calculate the motion of the bob, and illustrate the bob motion through real-time animation. The animation also indicates the relative height of the pendulum. This feature is valuable because the bob position cannot be observed by the student’s bare eyes from the real pendulum motion.

## 7. Conclusions

In this paper, we designed and implemented SimTalk, the simulation mechanism for an IoT application development platform called IoTtalk. We first elaborated on how to implement the simulator for an input IoT device (a sensor). We used a smart farm application to show how the simulator of sensors is used for correct implementation. Then we described how an output IoT device (an actuator) can be simulated by an animated simulator. We used applications including interactive art (skeleton art and water dance) and the pendulum physics experiment as examples to illustrate how IoT application behavior investigation can be achieved in SimTalk. A demonstration video for SimTalk is available in [29]. The source codes for developing IoT DA can be found in [30].

As the main outcome of this paper, the SimTalk simulation codes can be directly reused for real IoT applications and vice versa. Furthermore, SimTalk is nicely integrated with the tool BigraphTalk to formally verify the IoT application configuration. Such features have not been found in any IoT simulator in the world.

There are three directions for SimTalk future development:Enhancing BigraphTalk to provide better simulation code verification for SimTalk;Developing a distribution-fitting feature in SimTalk to replace a measured trace by a fit gamma distribution;Implementing parameter sweeping to automate the execution of SimTalk with various values.

## Figures and Tables

**Figure 1 sensors-20-02563-f001:**
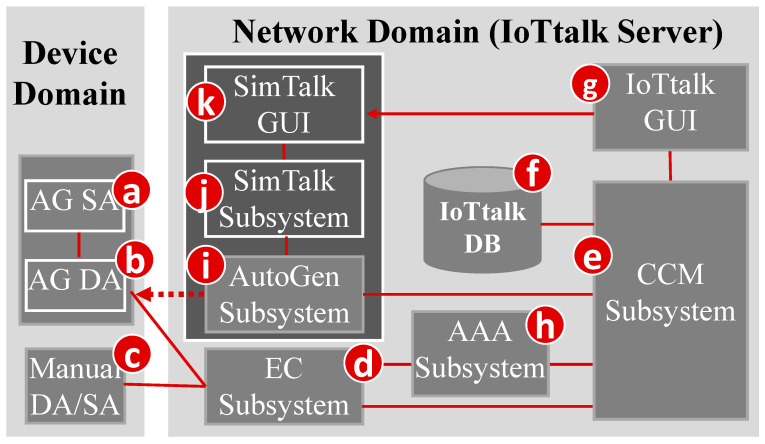
The IoTtalk architecture.

**Figure 2 sensors-20-02563-f002:**
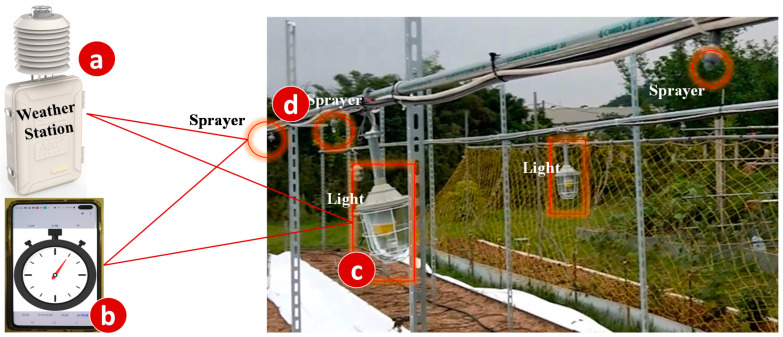
A simplified smart farm service.

**Figure 3 sensors-20-02563-f003:**
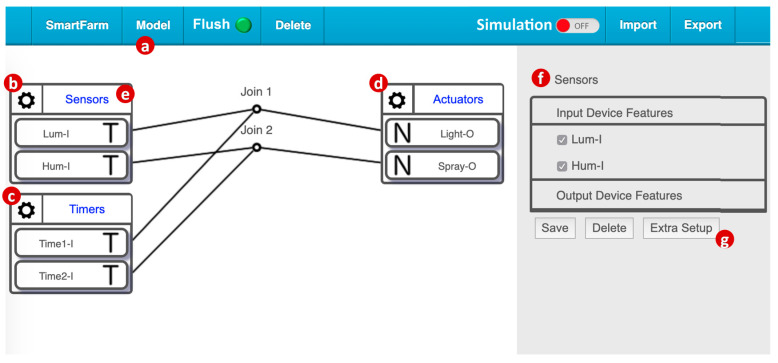
IoTtalk configuration for the smart farm service.

**Figure 4 sensors-20-02563-f004:**
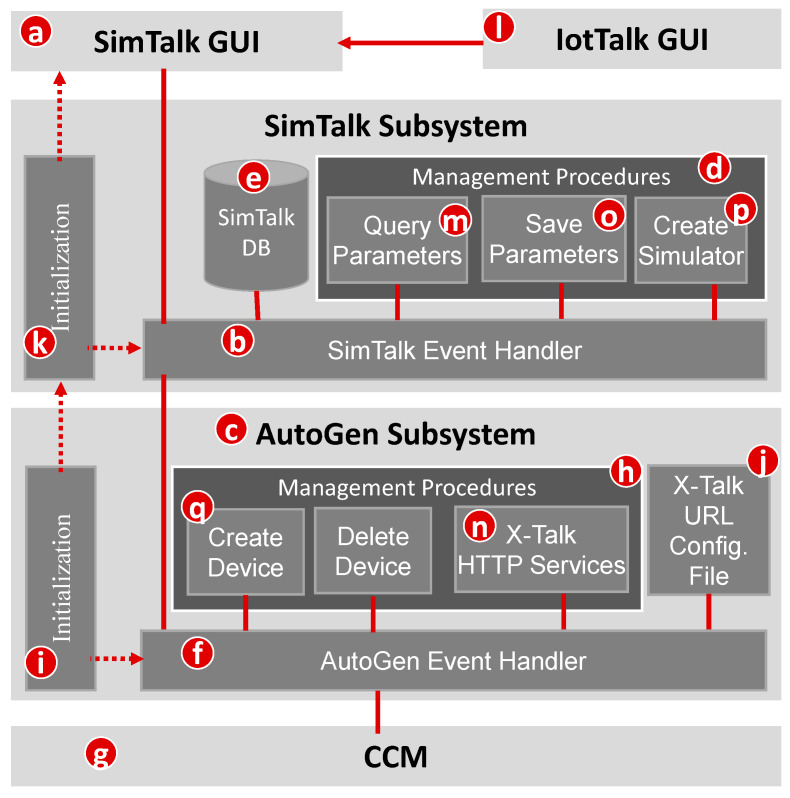
The SimTalk and the AutoGen Subsystems.

**Figure 5 sensors-20-02563-f005:**
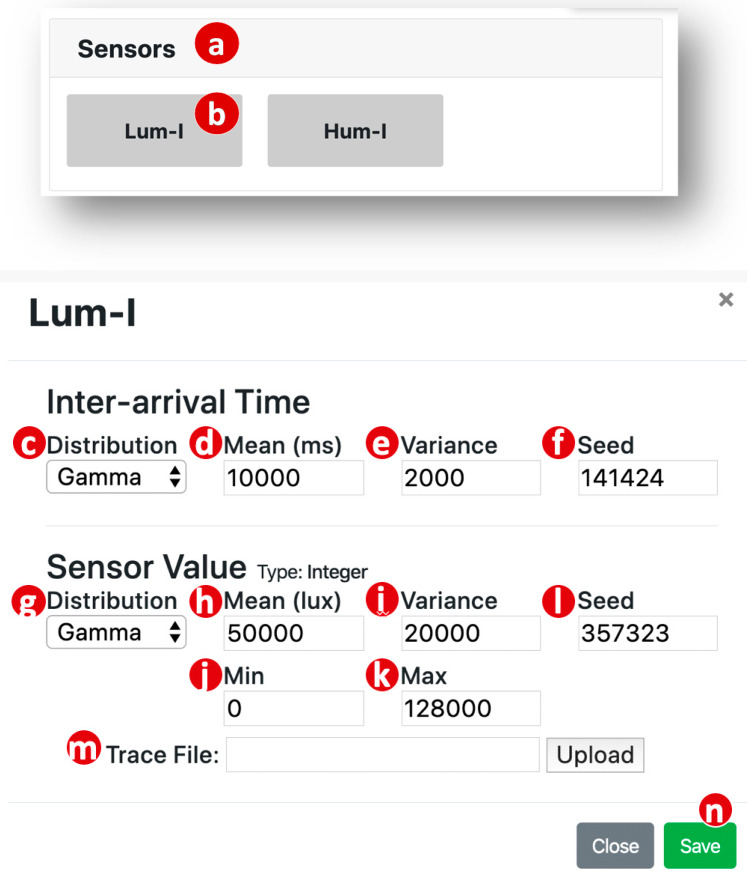
Web-based SimTalk graphical user interface (GUI).

**Figure 6 sensors-20-02563-f006:**
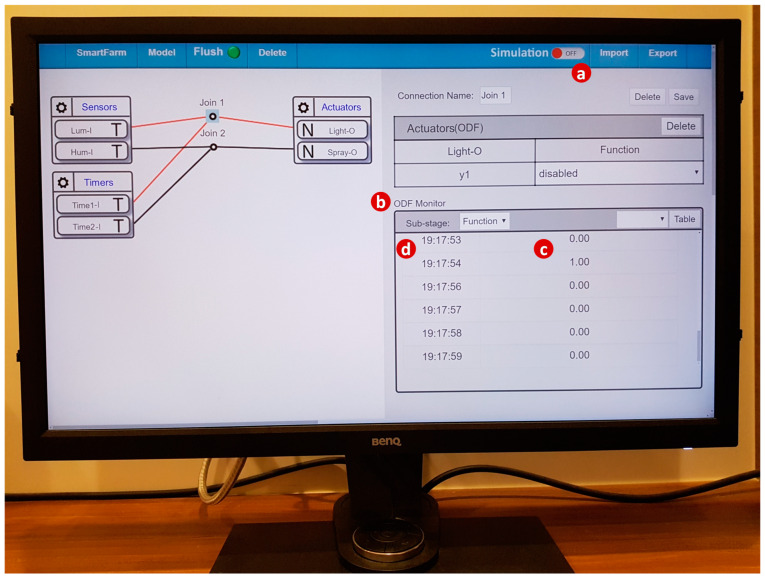
Web-based IoTtalk GUI (cont.).

**Figure 7 sensors-20-02563-f007:**
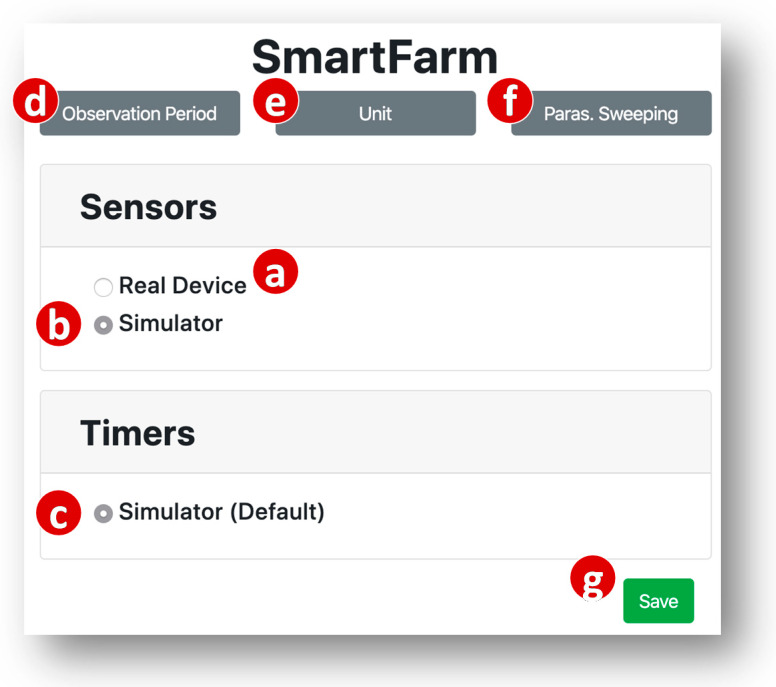
Web-based IoTtalk GUI (cont.).

**Figure 8 sensors-20-02563-f008:**
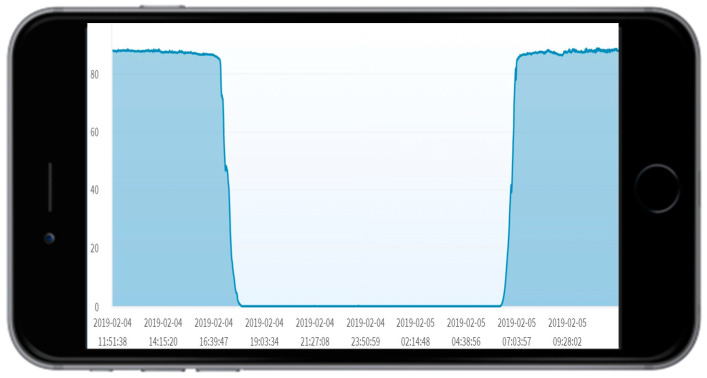
The time series chart of the output device feature (ODF) monitor (from Lum-I to Light-O).

**Figure 9 sensors-20-02563-f009:**
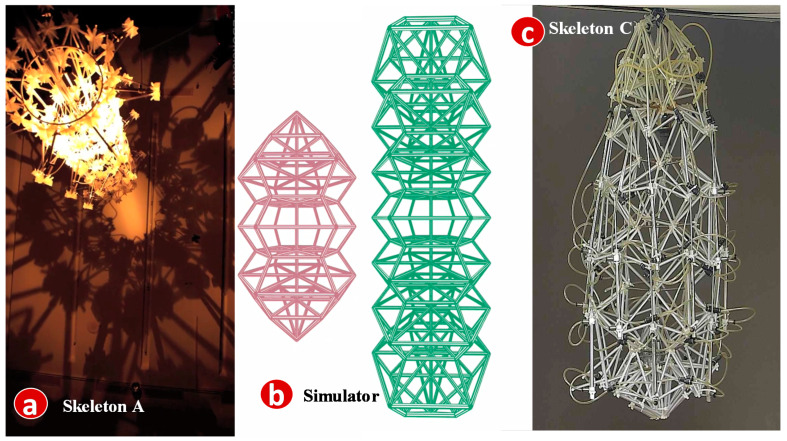
Skeleton: (**a**) geometric-shape shadow of skeleton; (**b**) shape change of the skeleton simulator; (**c**) another implementation of real skeleton device.

**Figure 10 sensors-20-02563-f010:**
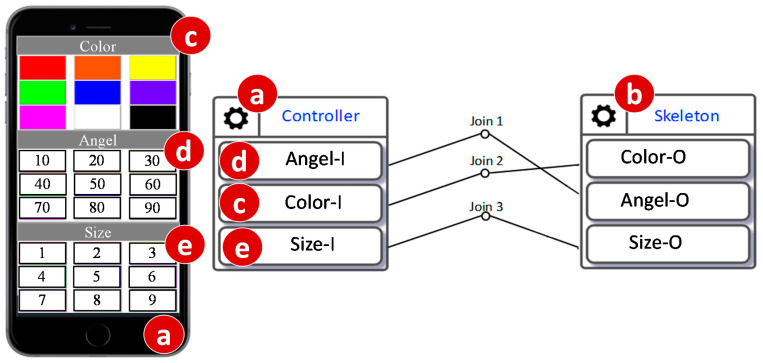
IoTtalk configuration for skeleton.

**Figure 11 sensors-20-02563-f011:**
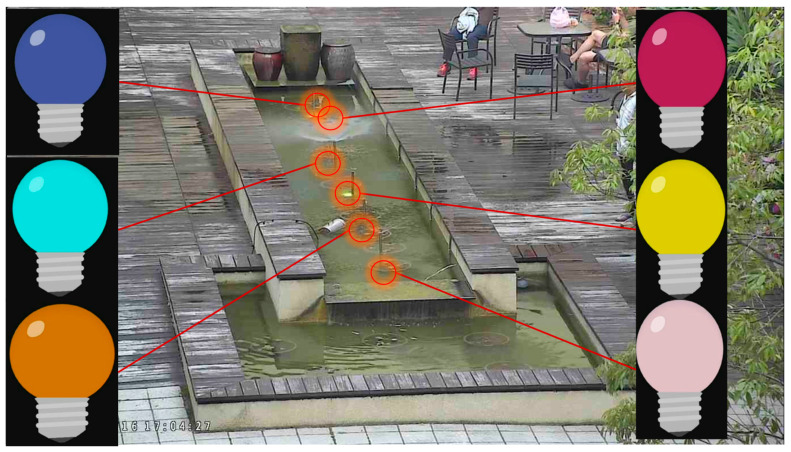
The real lights and the simulated lights in the water dance application.

**Figure 12 sensors-20-02563-f012:**
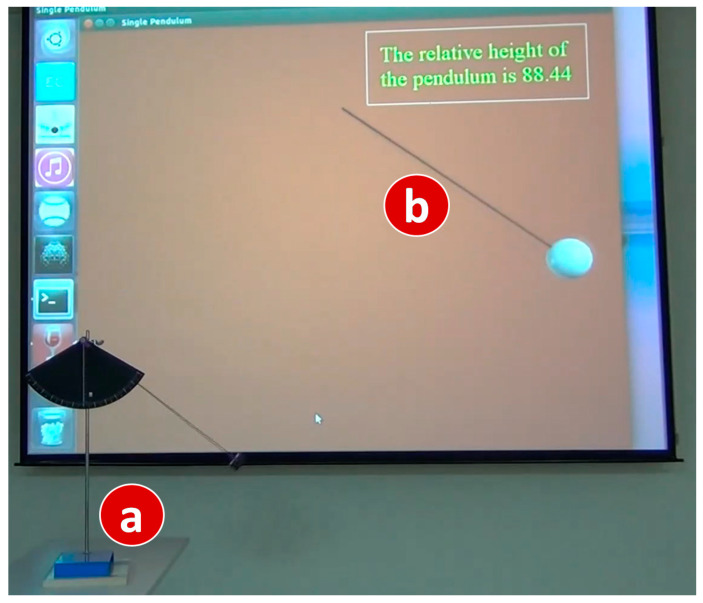
Simulation of pendulum.

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
