# Peer review of "SimTalk: Simulation of IoT Applications"

_sensors, 2020, doi:10.3390/s20092563_

Round 1

Reviewer 1 Report

This paper proposes a simulation platform, called SimTalk, which is designed  based on the IoTtalk, an IoT application development platform. The paper gives a notable contribution by giving a simulation tool for the researchers on IoT applications and services.    Before going to publication of the paper, the authors needs to consider  the following points.   1) It will be helpful if a comparison table is included by comparing the features of the existing ([11]~[14]) and proposed simulation tools (SimTalk) in Section 1 or 2.   2) The authors gives the demo video in [20]. It would be better if they also give the simulation source codes over a public website (e.g., GitHub or homepage).   3) Some typos may be checked. For example, in page 9, lines 281, the control sequences the lights and the sprinklers  -> the control sequences "of" the lights and the sprinklers.

Author Response

We would like to thank the reviewer for his/her constructive comments that have significantly improved the quality of our paper. We have addressed the reviewer comments point-by-point. Since we have included several figures in the revision report, the report is uploaded as the pdf file.

Reviewer 2 Report

The authors present SimTalk as an extension of the IoTtalk IoT application platform. SimTalk allows for the simulation of sensors and actuators for the design and testing of IoT systems, in case real input and output devices are not available or their use is not feasible. On the input side, probability distributions can be used to stochastically generate potential values of sensors. On the output side, animated simulators exist that can be used to visualize the behavior of actuators. Especially for testing whether a specific configuration of an IoT application results in the desired behavior, the presented tool seems valuable as it help the user to overcome the challenge of missing test environments. Furthermore, the existance of such tools is an important step towards the systematical and automated verification but also validation of IoT systems.

From a simulation point of view, the significance of the results generated by SimTalk could be improved. In the described functionalities of the application, the user can define the inter arrival time as well as the sensor values that will be generated by the simulated sensor. First, there is an option missing to define a seed value for the underlying random number generator. In case the behavior of the system is not as expected the user might want to replicate the exact same values multiple times to be able to compare the system's behavior before and after the modification. The use of seed values is also reaonable with respect to the replicability of results by the research community. Second, the user might have real values that were recorded from a real-world device. The process of applying distribution fitting approaches to convert such values into probability distributions is not trivial. This might be solved by adding the possibility to use real data or to enable users to parametrize distributions according to their need. A reference to a respective tool is provided below (Law 2011) In this regard, the displayed gamma distribution should not be defined via mean and variance but via shape parameter (k) and scale parameter (theta).

The authors do not elaborate on the execution of the simulation, i.e., the frequence of random number generation or the progress of time. Some sensors might send values in very short time-intervals whereas other sensors might only send values once in a while or when changes occur. Also, with respect to the systematic verification of the application, it might be valuable for the user of the tool be able to systematically test different values in the defined interval. A similar functionallity is for instance provided as part of the "BehaviorSpace" functionality of the NetLogo simulator (https://ccl.northwestern.edu/netlogo/).

Overall, the article is written in a hands-on style with strong emphasize on the GUI of the tools. The underlying concept of the presented components is not introduced neither how they will be integrated into the existing application. A more formal description of the concept would be beneficial.

The simulation of IoT systems and deviced is not a new field and multiple simulators for IoT systems have been presented. To provide a more comprehensive overview of this research field, the discussion of related work as well as the state of the art in simulation of IoT systems and applications should be added to the article. To this end, the authors should clarify how their tool differs from existing simulators.

Law, Averill M. "How the ExpertFit distribution-fitting software can make your simulation models more valid." Proceedings of the 2011 Winter Simulation Conference (WSC). IEEE, 2011.

Author Response

(The authors gave the same response as above.)

Reviewer 3 Report

The authors elaborate on how to implement the simulator for an input IoT device i.e. a sensor).

Then they describe how an output IoT device i.e. an actuator, can be simulated by an animated simulator.

They use a smart farm application to show how simulators of sensors are used for correct implementation.

The authors use applications including interactive art i.e skeleton art and water dance, and pendulum physics experiment as examples to illustrate how IoT application behavior investigation can be achieved in SimTalk.

=> The form of the presentation is not yet sufficiently prepared to be published.
=> Extensive editing of English language and style required
=>Comparisons with existing research in the field are sorely lacking in this study even if the authors cite up-to-date references

Author Response

(The authors gave the same response as above.)

Reviewer 4 Report

The paper is interesting and deals with a contemporary topic. The proposal is technically sound and the overall style fluent, but the manuscript looks like a technical report, then it requires a deep review. First of all, Authors should reserve more space for the paper's motivation in Introduction and for the presentation of the proposal's context. Then, they should insert a new section "Related Work" in which discuss the state-of-the-art about simulation (and simulator) in IoT and Cyberphysical systems. In such a new section, Authors should move lines 27-35 and survey important related works like  <Fortino, Giancarlo, Wilma Russo, and Claudio Savaglio. "Agent-oriented modeling and simulation of IoT networks." 2016 federated conference on computer science and information systems (FedCSIS). IEEE, 2016> <Zeng, Xuezhi, et al. "IOTSim: A simulator for analysing IoT applications." Journal of Systems Architecture 72 (2017): 93-107> <Wehner, Philipp, and Diana Göhringer. "Internet of things simulation using omnet++ and hardware in the loop." Components and Services for IoT Platforms. Springer, Cham, 2017. 77-87> <Zeng, Xuezhi, et al. "IOTSim: A simulator for analysing IoT applications." Journal of Systems Architecture 72 (2017): 93-107> <Lera, Isaac, Carlos Guerrero, and Carlos Juiz. "YAFS: A simulator for IoT scenarios in fog computing." IEEE Access 7 (2019): 91745-91758> <Chernyshev, Maxim, et al. "Internet of things (iot): Research, simulators, and testbeds." IEEE Internet of Things Journal 5.3 (2017): 1637-1647>  <D'Angelo, Gabriele, Stefano Ferretti, and Vittorio Ghini. "Simulation of the Internet of Things." 2016 International Conference on High Performance Computing & Simulation (HPCS). IEEE, 2016>, just to name a few. The relationship between the proposal and past authors works should be also better clarified. Conclusion should be also reinforced with a critical analysis of the main outcome of the work and future research line outlined. Finally, Authors should proof-read the manuscript looking for minors, typos and double-check the figure alignment (typically centered).

Summarizing, the paper has merit but needs also some major interventions.

Author Response

(The authors gave the same response as above.)

Round 2

Reviewer 3 Report

After the corrections made by the authors, I accept the article in this form. 

Reviewer 4 Report

Authors provided an enhanced version of the paper which can be accepted as long as a final proof-read (targeted at finding minors).